# Analysis of Cell Immunity for Children Infected with SARS-CoV-2 and Those Vaccinated against SARS-CoV-2 Using T-SPOT^®^.COVID

**DOI:** 10.3390/microorganisms12050975

**Published:** 2024-05-13

**Authors:** Tomohiro Oishi, Yuto Yasui, Atsushi Kato, Satoko Ogita, Takahiro Eitoku, Hideo Enoki, Takashi Nakano

**Affiliations:** Department of Clinical Infectious Diseases, Kawasaki Medical School, 577, Matsushima, Kurashiki 701-0192, Okayama, Japan

**Keywords:** cell immunity, COVID-19, children, T-SPOT^®^

## Abstract

Cellular immunity is critical for the regulation of viral diseases, including coronavirus disease 2019 (COVID-19), and is generally considered immature in childhood. However, the details of cellular immunity against severe acute respiratory syndrome coronavirus 2 (SARS-CoV-2) infection among children are unclear. We assessed cellular immunity in eight children post-vaccination against SARS-CoV-2 and 11 children after SARS-CoV-2 infection using the T-SPOT^®^.COVID assay for the spike (S) and nucleocapsid (N) proteins. In the vaccinated group, the T-SPOT^®^.COVID assay for the S protein yielded positive results in seven children. In the post-infection group, the assay for the N protein was positive for 5 of 11 children, with 3 of these 5 children requiring hospitalization, including 2 who needed mechanical ventilation. The T-SPOT^®^.COVID assay is thus valuable for assessing cellular immunity against SARS-CoV-2, and most children infected with SARS-CoV-2 may not develop such immunity unless the disease severity is significant.

## 1. Introduction

Immune responses, including humoral immunity and cell immunity against severe acute respiratory syndrome coronavirus 2 (SARS-CoV-2), are induced by vaccinations against SARS-CoV-2 or by SARS-CoV-2 infection [1,2]. In humoral immunity, antibody neutralization sensitivity reportedly decreases for the SARS-CoV-2 variant [3]. Cellular immunity is effective against new SARS-CoV-2 variants [4]. However, unlike humoral immunity, which involves antibodies, measuring cellular immunity is challenging owing to the large blood sample volume and the immediate processing needed for lymphoid cell separations. This is particularly difficult in pediatric patients. Therefore, only a few reports have been published on cell immunity against SARS-CoV-2 for children infected with them [5,6].

A commercial kit, namely, T-SPOT^®^.COVID (Oxford Immunotec, Oxfordshire, UK), may be useful to evaluate cellular immunity against SARS-CoV-2 because it can be used in several medical settings [7]. Although an evaluation of cellular immunity against SARS-CoV-2 using this kit for adults has been reported [8], no studies have investigated its use in children. Therefore, in this study, we aimed to analyze the cellular immunity against SARS-CoV-2 in children using the T-SPOT^®^. kit to elucidate the relationship between cellular and humoral immunity following coronavirus disease 2019 (COVID-19) vaccination or diagnosis. 

## 2. Materials and Methods

### 2.1. Ethics Aspects

Informed consent was obtained from children or their parents. The study protocol was approved by the ethics committee of Kawasaki Medical School, Kurashiki, Japan, on 1 August 2023 (no. 5370-04).

### 2.2. Sample Collection

In our study, we analyzed blood samples isolated from children vaccinated for COVID-19 or with prior COVID-19 diagnosis. These samples were collected from children at the children’s ward or pediatric clinic of Kawasaki Medical School Hospital. We recorded the age, sex, medical histories, frequencies of COVID-19 diagnosis or vaccination, and the time elapsed since COVID-19 diagnosis or vaccination.

### 2.3. SARS-CoV-2 Antigen-Specific T-Cell Responses 

We assessed SARS-CoV-2 antigen-specific T-cell responses after infection and vaccination for COVID-19 using the T-SPOT^®^.COVID kit (Oxford Immunotec, Oxfordshire, UK) [7]. The assay was performed exclusively using materials from the kit following the manufacturer’s instructions. Specifically, on day 1, the following stimulators were added in a volume of 50 μL per well: AIM-V as a negative control, phytohemagglutinin as a positive control, and SARS-CoV-2 spike subunit 1, nucleocapsid protein, and membrane protein peptide pools that exclude peptide sequences homologous to endemic coronaviruses. Briefly, 2.5 × 10^5^ peripheral blood mononuclear cells (PBMCs) in 100 μL AIM-V media were added to each well. The microtiter plate was incubated for 16–20 h at 37 °C with 5% CO_2_ in a humidified atmosphere. On day 2, cells were washed using PBS, and an alkaline phosphatase-conjugated antibody was added to the wells. Cells were incubated for one hour at 7 °C. The cells were washed with PBS, and the substrate was added to the wells and incubated at room temperature for 7 min. Subsequently, the reaction was terminated with demineralized water. Positive and negative controls for this assay were set.

These assays identify SARS-CoV-2 antigen-specific interferon-gamma-secreting T cells, primarily consisting of CD4+ T helper type 1 cells and CD8+ cytotoxic T cells, essential for an antiviral immune response. PBMCs were stimulated with overlapping peptide pools of SARS-CoV-2 spike subunit 1 (S1), nucleocapsid protein (N) in these assays. Peptides highly homologous to endemic coronaviruses were omitted from the peptide pool for S1 and N in T-SPOT^®^.COVID to minimize the chance of T-SPOT^®^.COVID responses being elicited by cross-reactive T cells in the absence of SARS-CoV-2 infection [8]. Blood samples for the T-SPOT^®^.COVID tests were drawn into heparin tubes and subsequently shipped to LSI Medience Corporation (Tokyo, Japan) in temperature-regulated boxes.

The T-SPOT^®^.COVID tests were processed and analyzed following the manufacturer’s instructions.

### 2.4. Serum Immunoglobulin (Ig) G and Enzyme-Linked Immunosorbent Assay for SARS-CoV-2

Anti-SARS-CoV-2 antibody levels were measured using the Elecsys^®^ Anti-SARS-CoV-2 RUO^®^ assay (Roche Diagnostics, Basel, Switzerland), which is based on the modified double-antigen sandwich immunoassay with recombinant protein representing the nucleocapsid antigen that measures the total antibody against SARS-CoV-2 (pan immunoglobulin). Specifically, total antibody detection was based on double-antigens sandwich immunoassay, and the IgM antibody detection was based on μ-chain capture immunoassay. Mammalian cell-expressed recombinant antigens containing the receptor binding domain of the SARS-CoV-2 spike protein were used to develop total antibody and IgM antibody assays. IgG antibody kits were used as indirect immunoassays, and a recombinant nucleoprotein of SARS-CoV-2 expressed in *Escherichia coli* was used as a coating antigen. We set positive and negative controls for this assay.

The assay was performed on a fully automated Cobas e801 analyzer (Roche Diagnostics) as described in a previous study [9].

## 3. Results

Table 1 presents data on demographics, T-SPOT^®^.COVID, and IgG for SARS-CoV-2 in eight children were vaccinated against COVID-19. The mean ± standard deviation of children’s age was 13 ± 2.9 after COVID-19 vaccinations and 9 ± 4.0 in those infected with SARS-CoV-2. All of them were vaccinated with the messenger ribonucleic acid (mRNA)-based vaccine of Pfizer BioNTech. They were vaccinated at fixed schedules, that is, two doses, 0.2 mL/dose, at intervals of 21–28 days. 

Their ages ranged from 7 to 15 years, comprising six males and two females. The interval from the last vaccination to sample collection varied between 9 days and 4 months, with vaccination frequencies ranging from 1 to 3. Two participants were diagnosed with COVID-19 before sample collection. No child showed an immunodeficiency.

T-SPOT^®^.COVID targeting S proteins was positive in seven of the eight children; the remaining case was not determined. In contrast, the T-SPOT^®^.COVID targeting N proteins was positive in only one individual who had a history of COVID-19. 

Data from the 11 children with a history of COVID-19 are shown in Table 2. Their ages ranged from 1 to 14 years, and there were four males and seven females. The duration from the onset of COVID-19 to sample collection ranged from 3 days to 4 months. Two individuals had been vaccinated before contracting COVID-19. Five were admitted to the hospital, two of whom required mechanical ventilation. 

The T-SPOT^®^.COVID targeting S proteins yielded positive results in 4 of 11 children, indeterminate results in 2, and negative results in 5 cases. Regarding the T-SPOT^®^.COVID targeting N proteins, the results were positive in three cases (all of whom were also positive for S proteins), indeterminate in three, and negative in five. Among the seven children, eight tests for SARS-CoV-2 IgG against both S and N proteins were conducted: five tests each were positive for IgG against S and N proteins, respectively. Three tests were negative for both.

## 4. Discussion

We analyzed cell immunity against SARS-CoV-2 among children using the T-SPOT^®^.COVID kit along with humoral immunity against SARS-CoV-2. This is the first study to evaluate cell immunity against SARS-CoV-2 after COVID-19 vaccination and diagnosis among healthy children using the T-SPOT^®^.COVID.

Approximately all children [87.5%] vaccinated against COVID-19 tested positive for T-SPOT^®^.COVID targeting S protein. In adults, cellular immunity against SARS-CoV-2 persists long after vaccination [10]. Similarly, we demonstrate that children can acquire cellular immunity against SARS-CoV-2 upon vaccination. However, many children tested for COVID-19 infection using T-SPOT^®^.COVID, which targets both the S and N proteins. We hypothesized that the limited positive cellular immunity observed in children post-infection in our study may be attributed to the mild severity of COVID-19 in children. In most cases, children with SARS-CoV-2 experience mild symptoms [11]. Primed innate immunity among children is useful for fighting SARS-CoV-2 infection [12]. Therefore, cell immunity might be unnecessary among many children with mild cases of COVID-19. Notably, two hospitalized children who needed mechanical ventilation, indicating severe COVID-19 cases, exhibited positive results for T-SPOT^®^.COVID targeting the S and N proteins. A previous report indicated that in children with SARS-CoV-2, the innate immunity response was higher than in adults, showing an inverse correlation with the severity of COVID-19 [13]. This knowledge lends support to our hypothesis. Therefore, we believe that although children can acquire cellular immunity against SARS-CoV-2 via vaccination, this immunity may not be adequately developed in many cases of infection.

We analyzed humoral immunity against SARS-CoV-2. Because all children after COVID-19 vaccinations developed SARS-CoV-2 IgG against S proteins, COVID-19 vaccinations in this age group are assumed to be useful. SARS-CoV-2 IgG against N proteins was positive in three of them, and the history of COVID-19 was unclear in two cases. Therefore, asymptomatic COVID-19 might occur in children. Among children who had a COVID-19 history, some were positive for SARS-CoV-2 IgG even when the T-SPOT^®^.COVID result was negative. This discrepancy is assumed to be caused by insufficient cellular immune responses in children, because of either the immaturity of cellular immunity or the severity of COVID-19. Regarding the relationship between the severity of COVID-19 and immunity, cellular immunity levels against COVID-19 were higher in adults with severe COVID-19 than in those with milder cases. In contrast, in our study, children hospitalized because of COVID-19, indicating more severe cases, tended to have a higher positive rate of T-SPOT^®^.COVID than those who were not hospitalized. A previous study reported that early nasal mucosal immune response as a type of natural immunity was higher in children than in adults, which was inversely correlated with the severity of COVID-19 [13]. Therefore, the nasal mucosal immune response may be heightened, potentially reducing the need for robust cellular immunity in children who did not require hospitalization.

This study has some limitations. First, the sample size was small. Regrettably, collecting samples for cellular immunity studies in children poses a challenge, as more than 5 mL of blood is required, which further necessitates immediate separation of lymphoid cells. Therefore, our cellular immunity samples from children are considered valuable. Second, cellular immunity is generally considered to be immature in childhood [14], which might affect the responsiveness of the T-SPOT^®^.COVID test in children. Interferon-gamma release assays, including the T-SPOT^®^ test, for diagnosing latent tuberculosis, have uncertain efficacy in children owing to the immaturity of cellular immunity [15]. Nevertheless, in our study, the T-SPOT^®^.COVID test results were positive among children post-COVID-19 vaccination, suggesting that this kit may be effective for children. Third, we performed two assays, one for humoral immunity and the other for cellular immunity. These two types for COVID-19 show different courses. Humoral immunity for SARS-CoV-2 decreased within several months, but cell immunity for them continued for more than half a year after the COVID-19 vaccinations [16]. However, we could not follow up with participants for more than 6 months, which is not long. Therefore, it is imperative to monitor children following COVID-19 vaccinations or diagnosis for more than half a year. Finally, although the kit is useful, as described below, we did not use other methods or kits to evaluate the cellular immunity of children vaccinated for SARS-CoV-2 or those infected with SARS-CoV-2. Therefore, a comparison of this kit with the other ones should be performed to assess its effectiveness. 

## 5. Conclusions

We investigated cellular immunity for COVID-19 in children using T-SPOT^®^.COVID. Our findings affirmed the utility of this kit in assessing cellular immunity in pediatric populations. Children develop cellular immunity against SARS-CoV-2 upon vaccination; however, acquisition of immunity may be limited in those affected with COVID-19 infection, particularly if their illness is not severe. We anticipate that future studies will provide further insights into cellular immunity for COVID-19 among children, contributing to a deeper understanding of this important aspect of pediatric health.

## Figures and Tables

**Table 1 microorganisms-12-00975-t001:** Children after COVID-19 vaccinations (April–December 2022).

Age (Year-Old)	Sex	Date of Collection (Month/Year)	Date of Last Vaccination	Frequency of Vaccination	Onset of COVID-19	T-SPOT^®^. COVID Targeting S Proteins (Spot) *1	T-SPOT^®^. COVID Targeting N Proteins (Spot) *1	SARS-CoV-2 IgG to S Proteins (U/mL) *2	SARS-CoV-2 IgG to N Proteins (U/mL) *2	Note
7	M	May/2022	3 weeks prior	1	-	0 (−)	0 (−)	19.6 (+)	<0.8 (−)	
		August/2022	2 months prior	2	3 weeks prior	40 (+)	14 (+)	570 (+)	6.2 (+)	
11	M	June/2022	9 days prior	1	-	35 (+)	4 (−)	78.5 (+)	16.0 (+)	Close contact 1 month prior
		July/2022	1 month prior	3	-	32 (+)	7 (±)	71.3 (+)	17.6 (+)	
9	M	July/2022	2 weeks prior	2	-	≥50 (+)	4 (−)	1760 (+)	38.0 (+)	
13	F	June/2022	2 weeks prior	3	-	22 (+)	0 (−)	-	-	
		September/2022	3 months prior	3	-	11 (+)	1 (−)	1490 (+)	<0.8 (−)	
15	M	August/2022	4 months prior	3	-	7 (±)	0 (−)	8330 (+)	<0.1 (−)	
13	M	August/2022	2 months prior	3	-	23 (+)	−1 (−)	999 (+)	<0.8 (−)	
14	M	August/2022	4 months prior	3	2 months prior	12 (+)	3 (−)	11,000 (+)	3.9 (+)	
15	M	November/2022	2 months prior	3	-	41 (+)	1 (−)	542 (+)	<0.8 (−)	
Mean ± SD: 13 ± 2.9	M: 6 F: 2	May–November/2022	3 days prior –4 months prior	1–3	Positive: 9/11	Positive: 9/11 Equivocal: 1/11 Negative: 1/11	Positive: 1/11 Equivocal: 1/11 Negative: 9/11	Positive: 10/10 Equivocal: 0/10 Negative: 0/10	Positive: 5/8 Equivocal: 0/8 Negative: 3/8	

*1: ≥8 spots, positive (+); 5–7 spots, not determined (±); ≤4 spots, negative (−) by using the T-SPOT^®^.COVID kit. *2: ≥0.8 U/mL, positive (+); <0.8 U/mL, negative (−) by using the Elecsys^®^ Anti-SARS-CoV-2 RUO^®^ assay. SARS-CoV-2, Severe Acute Respiratory Syndrome Coronavirus 2; IgG, Immunoglobulin G; S, S proteins; N, N proteins; F, Female; M, Male; SD, standard deviation.

**Table 2 microorganisms-12-00975-t002:** Children infected with SARS-CoV-2 (April–December 2022).

Age (Year-Old)	Sex	Date of Collection (Month/Year)	Onset of COVID-19	Date of Last Vaccination	T-SPOT^®^. COVID Targeting S Proteins (Spot) *1	T-SPOT^®^. COVID Targeting N Proteins (Spot) *1	SARS-CoV-2 IgG to S Proteins (U/mL) *2	SARS-CoV-2 IgG to N Proteins (U/mL) *2	Note
6	F	May/2022	5 days prior	-	2 (−)	4 (−)	<0.8 (−)	<0.8 (−)	Hospitalization
		June/2022	13 days prior	-	5 (±)	5(±)	-	-	
4	M	July/2022	3 days prior	-	0 (−)	0 (−)	<0.8 (−)	<0.8 (−)	Hospitalization
		July/2022 *1	3 weeks prior	-	2 (−)	3 (−)	4.3 (+)	1.4 (+)	
11	F	July/2022	4 months prior	-	2 (−)	0 (−)	1.2 (+)	<0.8 (−)	
9	F	July/2022	4 days prior	-	6 (±)	7 (±)	-	-	Hospitalization
		August/2022	2 weeks prior	2 months prior	40 (+)	14 (+)	570 (+)	6.2 (+)	
7	M	August/2022	2 months prior	5 months prior	12 (+)	3 (−)	11,000 (+)	3.9 (+)	
14	M	August/2022	2 weeks prior	-	19 (+)	27 (+)	5.32 (+)	4.6 (+)	Mechanical ventilation
3	F	August/2022	12 days prior	-	20 (+)	30 (+)	<0.8 (−)	2.5 (+)	Mechanical ventilation
1	F	September/2022	1 month prior	-	3 (−)	3 (−)	-	-	
11	M	September/2022	3 weeks prior	-	3 (−)	3 (−)	-	-	
11	F	December/2022	1 month prior	-	2 (−)	7 (±)	-	-	
9	F	May/2022	5 days prior	-	2 (−)	4 (−)	<0.8 (−)	<0.8 (−)	
Mean ± SD: 9 ± 4.0	M: 4 F: 7	May–November/2022	3 days prior to 4 months prior		Positive: 4/13 Equivocal: 1/13 Negative: 8/13	Positive: 2/13 Equivocal: 3/13 Negative: 8/11	Positive: 5/8 Equivocal: 0/8 Negative: 3/8	Positive: 5/8 Equivocal: 0/8 Negative: 3/8	

*1: ≥8 spots, positive (+); 5–7 spots, not determined (±); ≤4 spots, negative (−) by using the T-SPOT^®^.COVID kit.*2: ≥0.8 U/mL, positive (+); <0.8 U/mL, negative (−) by using the Elecsys^®^ Anti-SARS-CoV-2 RUO^®^ assay. SARS-CoV-2, Severe Acute Respiratory Syndrome Coronavirus 2; IgG, Immunoglobulin G; F, Female; M, Male; S, S proteins; N, N proteins; SD, standard deviation.

## Data Availability

The raw data supporting the conclusions of this article will be made available by the corresponding author, Tomohiro Oishi on request.

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
