# Peer review of "Analysis of Cell Immunity for Children Infected with SARS-CoV-2 and Those Vaccinated against SARS-CoV-2 Using T-SPOT®.COVID"

_microorganisms, 2024, doi:10.3390/microorganisms12050975_

Round 1

Reviewer 1 Report

Comments and Suggestions for Authors

The communication about cellular immunity in infants after infection with SARS-CoV-2 using the commercial T-SPOT®.COVID kit has many flaws and lacks novelty.

-              Although the article is a communication, the abstract and introduction lack the information necessary to understand the study's rationale.

-              The methodology requires including whether the recruited patients have other pathologies that could interfere with the results, what type of vaccine they have, the age of the patients, etc.

-              The tables presented are not very explanatory and are difficult to follow to conclude from the results obtained.

-              The number of samples analyzed and the great diversity of cases make it challenging to analyze the possible advantages of using the kit.

-              The advantage of using this kit in comparison with the in-house in these samples must be demonstrated since, as stated, it is not very efficient when it is a mild disease.

-              Is the kit's effectiveness the same in stopping the cellular response induced for all virus variants?

-              Other articles analyze cell immunity using more informative techniques than the one presented.

Comments on the Quality of English Language

Minor editing of English language required.

Author Response

Dear reviewer.

Thank you for your kindly comments.

I revised my article as you indicated. Revised points are as below written in boldfaced and italic letters.

The communication about cellular immunity in infants after infection with SARS-CoV-2 using the commercial T-SPOT®.COVID kit has many flaws and lacks novelty.

             Although the article is a communication, the abstract and introduction lack the information necessary to understand the study's rationale.

Thank you for your comment, I agree that this article is categorized as a communication.

The methodology requires including whether the recruited patients have other pathologies that could interfere with the results, what type of vaccine they have, the age of the patients, etc.

Thank you for your questions. All of enrolled children were vaccinated the messenger ribonucleic acid (mRNA)-based vaccine of The Pfizer BioNTech ones. Therefore, I added the description on L78-79. And as far as we checked, there was no one with immunodeficiencies. Therefore, I also described it on L83-84.

The number of samples analyzed and the great diversity of cases make it challenging to analyze the possible advantages of using the kit.

Thank you. As you suggested, the number of samples are small as I described on L159-. Therefore, we should take more samples and analyze advantages of this kit.

The advantage of using this kit in comparison with the in-house in these samples must be demonstrated since, as stated, it is not very efficient when it is a mild disease.

Thank you. As you commented, this kit is possible not to be very efficient for children with mild of COVID-19 as we demonstrated in our article. However, I think it is important that this was proved thorough our data.

-              Is the kit's effectiveness the same in stopping the cellular response induced for all virus variants?

Thank you for your question. Because this kit is for cell-immunity for SARS-CoV2, the antigens which this kit respond do not relate with spike proteins of them. That is, this kit is used common antigen among all SARS-CoV2 variants.

-           

  Other articles analyze cell immunity using more informative techniques than the one presented.

Thank you. The analysis on cell immunity for COVID-19 are written mainly in study among adults in articles of Reference No. 7,10, and 12. However, this kit is easier technique than those in the past reports.

Reviewer 2 Report

Comments and Suggestions for Authors

The communication by Oishi et al. reported the analysis of cell immunity using T-SPOT®.COVID kit in a cohort of SARS-CoV-2 infected or vaccinated children. The paper could be interesting but needed to be deeply revised and improved by clarity and dept.

The English used is generally good, but I recommend a proofreading by a native English speaker to improve the clarity and readability of the paper.

Title

I suggest rephrasing the title and use the correct terminology: the infection is with the SARS-CoV-2 virus, the COVID-19 is the disease.

Abstract

-       I suggest to better structure the abstract.

-       It could be helpful for the reader to link the results to the conclusion. Now it is not clear why your results are important in the treatment and diagnosis of Febrile Neutropenia.

-       Please, add the aim of the study in the abstract.

Introduction

The introduction needs to be deeply revised: an improvement in clarity and concepts is strictly required.

-       Be consistent with the spelling of the commercial name of the kit: in the keywords list it is reported T-spot® while along the manuscript is reported as T-SPOT®.COVID

-       Line 19: “most children infected with COVID-19”. I suggest substituting COVID-19 with SARS-CoV-2 since COVID-19 is the disease, while the causative agent is the virus SARS-CoV-2.

-       Lines 23-25: I suggest rephrasing the sentence since it is not very clear. Be carful with the use of COVID-19 and SARS-CoV-2 as I suggested above: vaccination against SARS-CoV-2 and not COVID-19.

-       Line 32: Infected with SARS-CoV-2 and not with COVID-19!

-       Line 35: “in” before UK is not necessary.

Materials and Methods

-       What type of vaccination did the children got? How many doses?

Results

-       Table 1 and 2 are not very clear and difficult to read. I suggest, at least, to switch the page layout to landscape.

-       Lines 78 and 97: Add the mean ± standard deviation of children’s age.

-       I suggest adding histogram to compare the percentage of positivity obtained with T-SPOT®.COVID kit and the other methods (Elecsy and ELISA kit). It would be straight forward for the readers to understand the results rather than going through the tables.

Discussion

-       Line 116: I would change against COVID-19 with SARS-CoV-2.

-       Line 120: “Nearly…” add into brackets the number or the percentage you have found.

Comments on the Quality of English Language

The English used is generally good, but I recommend a proofreading by a native English speaker to improve the clarity and readability of the paper.

Author Response

Dear reviewer.

Thank you for your kindly comments.

I revised my article as you indicated. Revised points are as below written in boldfaced and italic letters.

Title

I suggest rephrasing the title and use the correct terminology: the infection is with the SARS-CoV-2 virus, the COVID-19 is the disease.

Thank you, I revised the title as you suggested.

Abstract

-       I suggest to better structure the abstract.

-       It could be helpful for the reader to link the results to the conclusion. Now it is not clear why your results are important in the treatment and diagnosis of Febrile Neutropenia.

-       Please, add the aim of the study in the abstract.

Thank you, I added the sentence on L11-13 as you suggested.

Introduction

The introduction needs to be deeply revised: an improvement in clarity and concepts is strictly required.

-       Be consistent with the spelling of the commercial name of the kit: in the keywords list it is reported T-spot® while along the manuscript is reported as T-SPOT®.COVID

Thank you, I revised the spelling in the keywords as you indicated.

-       Line 19: “most children infected with COVID-19”. I suggest substituting COVID-19 with SARS-CoV-2 since COVID-19 is the disease, while the causative agent is the virus SARS-CoV-2.

Thank you, I revised here as you suggested.

-       Lines 23-25: I suggest rephrasing the sentence since it is not very clear. Be careful with the use of COVID-19 and SARS-CoV-2 as I suggested above: vaccination against SARS-CoV-2 and not COVID-19.

Thank you, I revised here as you suggested.

-       Line 32: Infected with SARS-CoV-2 and not with COVID-19!

Thank you, I revised here as you suggested.

-       Line 35: “in” before UK is not necessary.

I am sorry to mistake.  I revised here as you suggested.

Materials and Methods

-       What type of vaccination did the children got? How many doses?

Thank you, I added the information you asked on L80-83.

Results

-       Table 1 and 2 are not very clear and difficult to read. I suggest, at least, to switch the page layout to landscape.

    I am sorry to difficult to read. I gathered each table in one page.

-       Lines 78 and 97: Add the mean ± standard deviation of children’s age.

  Thank you. I added this information on 80-82, Table 1, and Table 2.

-       I suggest adding histogram to compare the percentage of positivity obtained with T-SPOT®.COVID kit and the other methods (Elecsy and ELISA kit). It would be straight forward for the readers to understand the results rather than going through the tables.

   I agree with your suggestion. However, we did not compare T-SPOT®.COVID kit to the

other kit. Therefore, I added this point on L 185-189 as a limitation,

Discussion

-       Line 116: I would change against COVID-19 with SARS-CoV-2.

  Thank you. I changed “against COVID-19” to “SARS-CoV-2” as you suggested.

-       Line 120: “Nearly…” add into brackets the number or the percentage you have found.

Thank you. I added the number or the percentage into bracket.

Reviewer 3 Report

Comments and Suggestions for Authors

The authors should further exploit their results by comparing the different variables between the various groups (before vaccination, with COVID-19, and post-vaccination) through graphs and statistical analysis.

Author Response

Dear reviewer.

Thank you for your kindly comments.

I revised my article as you indicated. Revised points are as below written in boldfaced and italic letters.

The authors should further exploit their results by comparing the different variables between the various groups (before vaccination, with COVID-19, and post-vaccination) through graphs and statistical analysis.

 Thank you for your suggestion. However, we did not have chances to compare the different variables between the various group. Therefore, I would like to try more research as you indicated in the near future.

Round 2

Reviewer 1 Report

Comments and Suggestions for Authors

The authors should have considered my suggestions properly, and the article needs to be improved.

Comments on the Quality of English Language

 Minor editing of the English language is required.

Author Response

Response to Academiec editor uploaded.

Reviewer 2 Report

Comments and Suggestions for Authors

The authors responded satisfactorily to my requests and made all the corrections I suggested. I see no other problems in the manuscript. 

Therefore, I consider that the manuscript should be accepted for publication in its current form.

Author Response

Response to Academiec editor uploaded.

Reviewer 3 Report

Comments and Suggestions for Authors

None

Author Response

Response to Academiec editor uploaded.